# Machine Learning Algorithms Applied to Predict Autism Spectrum Disorder Based on Gut Microbiome Composition

**DOI:** 10.3390/biomedicines11102633

**Published:** 2023-09-26

**Authors:** Juan M. Olaguez-Gonzalez, Isaac Chairez, Luz Breton-Deval, Mariel Alfaro-Ponce

**Affiliations:** 1School of Engineering and Science, Tecnologico de Monterrey, Monterrey 64849, Mexico; jmolaguez@gmail.com (J.M.O.-G.); isaac.chairez@tec.mx (I.C.); 2Institute of Advanced Materials for Sustainable Manufacturing, Tecnologico de Monterrey, Monterrey 64849, Mexico; 3Instituto de Biotecnología, Universidad Nacional Autónoma de México, Cuernavaca 62210, Mexico; lzbreton@ibt.unam.mx; 4Consejo Nacional de Ciencia y Tecnologia, Mexico City 03940, Mexico

**Keywords:** autism, ASD, machine learning, artificial neural networks, microbiome, microbiota

## Abstract

The application of machine learning (ML) techniques stands as a reliable method for aiding in the diagnosis of complex diseases. Recent studies have related the composition of the gut microbiota to the presence of autism spectrum disorder (ASD), but until now, the results have been mostly contradictory. This work proposes using machine learning to study the gut microbiome composition and its role in the early diagnosis of ASD. We applied support vector machines (SVMs), artificial neural networks (ANNs), and random forest (RF) algorithms to classify subjects as neurotypical (NT) or having ASD, using published data on gut microbiome composition. Naive Bayes, k-nearest neighbors, ensemble learning, logistic regression, linear regression, and decision trees were also trained and validated; however, the ones presented showed the best performance and interpretability. All the ML methods were developed using the SAS Viya software platform. The microbiome’s composition was determined using 16S rRNA sequencing technology. The application of ML yielded a classification accuracy as high as 90%, with a sensitivity of 96.97% and specificity reaching 85.29%. In the case of the ANN model, no errors occurred when classifying NT subjects from the first dataset, indicating a significant classification outcome compared to traditional tests and data-based approaches. This approach was repeated with two datasets, one from the USA and the other from China, resulting in similar findings. The main predictors in the obtained models differ between the analyzed datasets. The most important predictors identified from the analyzed datasets are *Bacteroides*, *Lachnospira*, *Anaerobutyricum*, and *Ruminococcus torques*. Notably, among the predictors in each model, there is the presence of bacteria that are usually considered insignificant in the microbiome’s composition due to their low relative abundance. This outcome reinforces the conventional understanding of the microbiome’s influence on ASD development, where an imbalance in the composition of the microbiota can lead to disrupted host–microbiota homeostasis. Considering that several previous studies focused on the most abundant genera and neglected smaller (and frequently not statistically significant) microbial communities, the impact of such communities has been poorly analyzed. The ML-based models suggest that more research should focus on these less abundant microbes. A novel hypothesis explains the contradictory results in this field and advocates for more in-depth research to be conducted on variables that may not exhibit statistical significance. The obtained results seem to contribute to an explanation of the contradictory findings regarding ASD and its relation with gut microbiota composition. While some research correlates higher ratios of *Bacillota/Bacteroidota*, others find the opposite. These discrepancies are closely linked to the minority organisms in the microbiome’s composition, which may differ between populations but share similar metabolic functions. Therefore, the ratios of *Bacillota/Bacteroidota* regarding ASD may not be determinants in the manifestation of ASD.

## 1. Introduction

ML techniques are reliable approaches for aiding in the diagnosis of complex diseases [1]. However, these techniques are primarily used to identify correlations rather than establish causation between input–output information in systems with uncertain descriptions [2]. Often, ML outcomes are misinterpreted as indicating causation, which can lead to imprecise consequences when making decisions based solely on data observations [3]. While such correlations can provide fundamental insights into relationships explaining the evolution of various medical disorders and illnesses, they must be interpreted cautiously. Within the realm of potential applications of ML in medicine, particularly in addressing ASD, researchers have explored the correlation between the composition of the gut microbiome and ASD. The outcomes of these studies have offered valuable insights into the potential link between ASD and gut microbiome composition (GMC) [4]. However, note that the existing literature presents contradictions among reported results, particularly regarding the specific bacteria associated with the disease.

According to Namkung [5], a microbiome can be defined as the “composition of the bacterial community in a specific environment”. Therefore, a gut microbiome refers to the bacterial community residing within the human gut. Sabit et al. define the microbiome as the “complete set of genomes of the microbial community which lives in symbiosis with the host,” and distinguish “gut microbiota” as the bacteria living within the gut, excluding viruses, protozoa, fungi, and archaea [6]. The composition of the gut microbiome influences not only the function and development of the immune system but also plays a significant role in various health conditions [7]. There are indications that intestinal bacteria might not only be linked to ASD but also to other diseases associated with metabolic and pro-inflammatory disorders [8]. Additionally, they may play a role in conditions such as Parkinson’s disease, multiple sclerosis, rheumatoid arthritis, obesity, type 1 diabetes (T1D), and the risk of heart disease [9,10,11]. In 2012, Yatsunenko et al. [12] concluded that environmental factors, including geographical location, interact with cultural differences such as diet and exposure to diverse microbes, influencing the composition of the gut microbiome.

In 2015, Reddy et al. found, based on human and animal studies data, that altered gut microbiota (dysbiosis) could be associated with ASD [13]. However, the exact bacterial composition (including its temporal variations) that contributes to ASD is still unclear. As stated by Rajkomar et al. [14], a data-driven strategy has become a fundamental technology required for cases where the volume of process data exceeds human comprehension. This is especially relevant for new approaches that not only identify statistical relationships among variables but also learn “extremely complex relationships” from heterogeneous medical data, including hospital records, medical notes, examination results, medical images, sensor time series, genomic data, and laboratory results, among others. Given that ML methods have demonstrated the potential to match human experts in diagnosing certain diseases [15,16], we hypothesize that ML is likely to surpass our current capacity to identify causal variables linking ASD and GMC. This work represents a step toward testing this hypothesis.

### 1.1. Autism Spectrum Disorder and Gut Microbiota Composition

Usually, ASD is diagnosed using widely accepted tests [17,18,19,20]. However, the main causes of this syndrome remain unclear. While the increasing ratio of ASD among newborns since the mid-20th century is a known fact, the underlying root causes remain elusive. Approximately 30% of ASD cases can be attributed to genetic predisposition, encompassing factors such as monogenic and polygenic syndromes, aberrations on the long arm of chromosome 15, numerical and structural abnormalities of the sex chromosomes, variations in copy number, incomplete penetrance, and variable expression. Fragile X syndrome stands out as a significant contributor among these genetic factors. Nevertheless, the origins of the remaining 70% of cases remain unclear [4].

Trying to explain causation, previous studies have investigated a range of factors, going from TV or audiovisual media exposure during the first year of life [21] and extending up to genetic factors [22].

Among these lines of investigation, a burgeoning research area centers around the correlation between the composition of the gut microbiome and the manifestation of ASD, an approach substantiated by several supportive studies [23,24,25,26,27,28,29,30].

The earliest studies concerning the gut microbiome composition and its relationship with ASD manifestation indicated that children affected by the syndrome exhibited high levels of neurotoxic metabolites, primarily attributed to the overgrowth of *Clostridium*. This overgrowth was often linked to excessive antibiotic use. This hypothesis was initially based on the findings of an index case, where regressive autism coincided with antibiotic abuse. Notably, symptoms improved with vancomycin treatment, but upon discontinuation of the treatment, most children reverted to their initial condition [31]. It is worth noting that early studies employed fluorescence in situ hybridization techniques to identify bacteria in stool samples [32]. As research designs improved, more advanced techniques were adopted, with the majority of published studies utilizing 16S rRNA sequencing methods and a tendency to use shotgun sequencing.

In 2019, Sharon et al. transplanted gut microbiota from humans diagnosed with ASD and from other neurotypical subjects (NT) as controls into germ-free mice. This study discovered that colonization with ASD-associated microbiota alone is sufficient to induce hallmark autistic behaviors [33]. Subsequently, a study by Kang et al. demonstrated the efficacy of microbial transplant therapy in humans, showcasing its ability to reduce both ASD and GI symptoms in a longitudinal study [34].

In 2021, Fouquier et al. [35] found that location plays an important role in researching microbiome composition and its influence on ASD. They highlighted that site location has the most significant impact when comparing the beta diversity found in samples. Analyzing microbiome compositions and their impact on ASD using samples collected from multiple countries presents an even greater challenge, necessitating the application of modern advanced methodologies, including machine learning tools. They additionally trained an RF classifier using ML to uncover relationships among selected genera (41 chosen after feature selection) and their impact on ASD manifestation. They then compared these results with another RF classifier trained using amplicon sequence variants (ASV) and employing principal coordinates analysis to reduce the ASVs to 13 predictors. In both cases of feature selection, a variable was considered if its value was at least 0.01. They achieved a maximum classification accuracy of 86% by combining samples from two different sampling sites and using amplicon sequence variants (ASV) as inputs. When attempting to classify using solely the samples collected in Arizona, the accuracy reached 66%, while for those collected in Colorado, it was 60%. Both cases used a model trained solely with Arizona’s samples. This suggests that while location alone is not a strong predictor of ASD status, the combination of samples from both locations yields a more effective predictor. Nevertheless, as is the case with all studies published thus far, the authors emphasize that the “current generalizability to other cohorts is not understood,” indicating their caution in interpreting the causal relationship leading to their results.

None of the studies published up to now have been able to explain the reasons for the relationship between the human gut microbiome and ASD. Their results are mainly local and are valid only for the population living in the place where the samples were taken. For example, most researchers found that *Clostridium* has higher relative abundance in ASD subjects than in NT subjects. Nevertheless, contradictory results, at the phylum level, have been published in the case of the *Bacillota/Bacteroidota* ratio, with several authors [28,36,37] claiming that a big *Bacillota/Bacteroidota* ratio is found in people diagnosed with ASD. In comparison, other authors found the exact opposite results [38,39]. Other conflicting outcomes have arisen at the genus level with *Akkermansia, Dorea, Sutterella, Faecalibacterium*, and *Ruminococcus* [25,32,37,39,40,41,42].

Some researchers opted to use siblings or twins in their studies [25,40], while others avoided their inclusion due to concerns about potential bias in the results [27]. Given that finding females with ASD is more challenging than finding males [43], the majority of the subjects involved in these studies were males. Note that, apart from Luna et al. [27] and Williams et al. [44], most studies involved both males and females. However, this approach may overlook innate differences in GMC based on sex [45]. Another influential factor is that the gut microbiome composition of an individual changes over time [35]. This implies that even though an individual’s ASD status remains constant, their bacterial concentrations can vary.

Considering that there are diverse factors in each of those studies and that there is still a lack of rules for conducting them, we are not likely to be able to generalize results and make a solid conclusion about the actual relationship between gut microbiome composition and ASD manifestation that may lead to the development of specific treatments.

Given the aforementioned facts and the advice from some authors to shift the focus toward studying the byproducts of microbial fermentation and their metabolism in an attempt to elucidate the causes of ASD related with GI symptoms, there has been a growing number of publications adopting this new approach with outcomes similar to previous studies. In 2019, Nogay et al. [46] stated, “With the available information, it is not yet possible to develop a gut microbiota-based nutritional intervention to treat gastrointestinal symptoms for individuals with autism.” However, the option of utilizing ML and other computational techniques has not been fully explored yet. The intricacies of this problem could potentially be addressed through the application of modern artificial intelligence (AI) techniques, particularly ML. It is hoped that these techniques can effectively relate key factors and provide insights, if not transform the perspective, for addressing such challenges [47].

### 1.2. Contribution of This Work

Our study aims to address this significant gap in current research. The hypothesis guiding the development of our work was based on the conventional approach to studying diseases, which primarily involves the utilization of significance tests to establish causation and either accept or reject hypotheses [48,49,50]. However, this approach may inadvertently overlook crucial factors when investigating complex neurodevelopmental diseases. This oversight becomes particularly noteworthy when we acknowledge that even small quantities of specific substances can exert a substantial influence on health, particularly brain function [51,52,53].

The primary contribution of this study lies in the development of ML models for classifying subjects with ASD based on their gut microbiota composition. Additionally, this work aims to draw attention to the need for further research on bacteria that emerged as predictors for the models, often considered statistically insignificant variables in human GMC. These variables could potentially play a role in contributing to environmental factors associated with ASD. Through the assistance of ML tools, it was observed that the key predictors are not necessarily those variables with the highest relative abundance in the environment or the ones exhibiting the most statistical differences. This novel approach, implemented on two datasets from different countries, has the potential to reshape how research on ASD and microbiome composition is traditionally conducted. Moreover, this innovative approach encourages the utilization of ML classifiers to gain insights into the primary predictors of ASD. These insights could inform the development of targeted therapeutic interventions aimed at influencing microbiome compositions.

### 1.3. Machine Learning Tools

Although several other ML tools were trained and evaluated (naive Bayes, k-nearest neighborhood, ensemble learning, logistic regression, linear regression, and decision tree), the ML tools reported in this study are RF, SVM, and ANN (multilayer perceptron), because these were the models that provided the best classification power.

RF is an ML classifier that relies on the binary decision tree, which splits the samples into two child nodes while trying to maximize the variance explained by the dependent variable. RF allows the creation of an ordered by importance variable list according to the scores of each variable in how influential they are in building the forest [54]. An SVM is an ML classification technique whose primary purpose is to classify binary events. However, it has been used in multigroup classification [55], and the classification process consists of finding the optimum set of support vectors as separator hyperplanes to classify the observations [56]. The multilayer perceptron is an architecture of the ANN where the parameters that affect the network’s behavior are known as weights and biases. Each of these ML tools may offer a suitable method to establish the relationship between the human gut microbiome composition and ASD prevalence. According to this argument, each of these ML tools were evaluated within the SAS Viya software 2021.2.4. Notice that, in the three types of ML algorithms presented in this work, the classification performance was achieved by supervised training and that the results were repeated in MATLAB when using the interior point as an external function optimizer.

Figure 1 shows the multilayer perceptron architecture used in this study. The “Link” is the weight of the connections between two neurons; the “Neuron Weight” represents the bias of each neuron. Finally, the size of the neurons within the hidden layer is the relative importance of the variables in the model developed. The ML algorithm can find the values that optimize the classification capabilities of the network. If the performance of the ANN is not satisfactory, the parameters must be changed until a good performance is achieved without falling into overfitting.

## 2. Methodology for Testing the ML Tools in ASD Classification

The sequence data for the microbiome composition were downloaded from the GeneBank Sequence Read Archive with the accession number PRJNA578223 and are from the research conducted by Zou et al. [23]. These data can be found at https://www.ncbi.nlm.nih.gov/bioproject/PRJNA578223/, accessed on 1 March 2022. The first dataset consists of 96 fecal samples, 48 from ASD children (from 2 to 7 years old, average 5, average BMI = 17.4, 10 females and 38 males) and 48 from NT children (all at 48 months, no allergies, 24 females and 24 males, average BMI = 16.3). The high-quality 16S rRNA sequences were obtained from the 96 samples following high-throughput DNA sequencing. The total gut microbiota was determined through the phylogenetic and taxonomic assessments of the 16S rRNA V3–V4 regions.

The second dataset was downloaded from the NCBI Biosample Database with the accession number PRJNA589343, and is from the research conducted by Ding et al. [24]. These data can be found at https://www.ncbi.nlm.nih.gov/bioproject/PRJNA589343/, accessed on 1 March 2022. The second dataset comprised 127 fecal samples, 77 from ASD children and 50 from NT children. In total, 59 of the 77 ASD children and 39 of the 50 NT children were males. For identification of the gut microbiota, the V4 region of 16S rRNA genes were amplified from DNA isolated from fecal samples. For identification of the gut microbiota, the V4 region of 16S rRNA genes was amplified from DNA isolated from fecal samples. One of the 127 samples presented too many inconsistencies and was not used to develop the machine learning models.

After performing quality control analysis using FASTQC, the datasets were filtered for high-quality and non-chimeric sequences using DADA2, considering a cutoff point when the average quality score was 20 as modulated by the pipeline parameters. The resulting amplicon sequence variants (ASV) were subjected to taxonomic annotation using Vsearch against the Silva-132 database. Alpha rarefaction analysis was performed to verify a sufficient sequencing depth using qiime2 v2020, and the classification was taken until the genus level. The absolute counts were converted into relative abundances expressed as percentages of the total amount of bacteria. All the samples were normalized on a scale [0,1] before being considered as inputs to the proposed classifiers. For this task, the min–max scaling was used:(1)$$x^{\prime }=\frac{\left(x_{i}\right)-\min\left(x\right)}{\max(x)-\min(x)}$$

For each attribute, *x* represents a vector of values for a given genus, min(*x*) denotes the minimum value within the vector, max(*x*) signifies the maximum value within the vector, $x_{i}$ represents the value to be normalized, and $x^{\prime }$ indicates the resulting normalized value.

No distinct feature selection technique was employed in this study. Instead, all variables were input into an initial SVM classifier. The first SVM was utilized as a filter to identify the 20 most relevant features, which were then employed to construct the models discussed in this study. Subsequently, the models underwent fine-tuning by adding, removing, or altering features, with the aim of determining the point at which the highest overall accuracy of the model was achieved. This juncture was considered the optimal feature quantity for each model.

The analysis depth was chosen up to the genus level because this is where “controversial findings are more often reported” [23]. However, within Ding’s dataset, an additional analysis was conducted, extending the depth to the species level. This extension was performed to enable a comparison between our RF model and that presented by Ding et al. [24] in their publication. In instances where certain species appeared as sequential order predictors, they were aggregated into their respective genera. This approach aimed to provide a broader perspective on other important variables and facilitated a more comprehensive comparative analysis.

The data used in this study were derived from existing research, which had already performed statistical analyses using common metrics in the medical field. These analyses were not duplicated in our study; instead, they were utilized for comparison with the most relevant variables identified by our ML models. We evaluated our model’s performance using metrics based on the confusion matrix such as sensitivity, specificity, Youden’s index, and overall prediction accuracy. All data preprocessing, normalization, and ML model development procedures were conducted in SAS Viya, utilizing cloud-based GPUs. Our study leverages the advanced ML techniques available in the SAS suite, facilitating a comprehensive analysis and fair characterization of the proposed techniques. One of the primary techniques for visualizing the outcomes of an ML model is the confusion matrix. This matrix provides a concise representation of the predicted and actual classification outcomes for various data segments, offering a rapid assessment of the model’s performance, as well as insight into true or false positive and negative ratios [57].

### 2.1. Support Vector Machine Classifier

The SVM functions in a manner similar to linear discriminant analysis (LDA). It constructs a hyperplane or a set of hyperplanes to segregate feature vectors into distinct classes, much as LDA does. However, SVM selects the hyperplane that maintains the maximum distance from the nearest training samples. In line with Cover’s theorem, SVM identifies the hyperplane with the greatest margin by mapping input data into a higher-dimensional space. Cover’s theorem posits that when addressing a complex classification problem in a high-dimensional nonlinear space, the likelihood of achieving linear separability is higher compared to when dealing with a low-dimensional nonlinear space [58].

The SVM classifier for both datasets used the interior point as the optimization technique. The kernel function was set to be linear with a C penalty method, which adds a penalty for every misclassified prediction and a penalty value of 1 with a tolerance of ${1\;✕\;10}^{-6}$. For the first dataset, the norm of the most extended vector was 1.3201135306, and the inner product of weights was 20.717258244. The total number of SVs was 47. For the second dataset, the norm of the most extended vector was 1.8244936022, and the inner product of weights was 19.690739291 while the total number of SVs was 76. In both cases, an SVM was the algorithm used to find the main predictors.

### 2.2. Random Forest Classifier

RF is a supervised classification algorithm that can find complex relationships among high-dimensional data commonly found in biological systems. The RF is formed from individual binary decision trees that split their decisions into two children to maximize the variance explained by the dependent variable [54]. RF for both datasets was run starting from 5 trees to 100 trees. The validation and testing misclassification ratios were recorded to find the best performer. Cutoff values range from 0 to 0.99, inclusive, in increments of 0.01. At each cutoff value, the predicted response classification is determined by whether the expected probability of the response diagnosis being NT is greater than or equal to the cutoff value. When the expected probability of the event is greater than or equal to the cutoff value, then the predicted classification is NT; otherwise, it is ASD.

### 2.3. Artificial Neural Network Classifier

ANNs have biologically inspired ML algorithms to simulate how the human brain processes information. They can find complex patterns through experience gained with a series of decision/feedback cycles [59].

For the first dataset, a single ANN was used with the following characteristics:The chosen architecture was the multilayer perceptron.The total number of neurons was 32.Twenty of them were used as inputs (one for each of the 20 genera used as predictors), two were used as output neurons, one for ASD, and the other for NT, while there were 10 hidden neurons in a unique hidden layer.

There were 210 weight parameters (WP) and 12 bias parameters (BP), although their actual positioning and values were not generated in the report offered by the SAS Viya software 2021.2.4 used for the modeling. For the second analysis, the ANN was composed of 20 input neurons, 18 hidden neurons in a single layer, and two output neurons, 318 WP and 20 BP, with a multilayer perceptron architecture for the second dataset. The actual connections and values for WP and BP were not generated in the report offered by the SAS Viya software 2021.2.4 used for the modeling.

### 2.4. Performance Metrics

As performance metrics, we compute the confusion matrix, which consists of values representing correct and incorrect classifications within the test partition. From this, we derive key indicators such as specificity (Sp), sensitivity (Sn) (also known as *recall*), overall accuracy (Acc), and the Youden’s index.

**Sp** is based on the true negative rate prediction; in our case, the number of actual NT cases within the test partition is correctly classified as NT,
(2)$$\mathrm{Sp}=\frac{TN}{TN+FP},$$
where TN stands for *true negative* and FP for *false positive*. This metric evaluates the model’s effectiveness in recognizing cases that do not present the disease.**Sn** is based on the true positive rate prediction; in our case, the number of actual ASD cases within the test partition is correctly classified as ASD,
(3)$$\mathrm{Sn}=\frac{TP}{TP+FN},$$
where TP stands for *true positive* and FN for *false negative*. This metric measures the capability to identify the cases that present a condition and is penalized when a false positive is predicted.**Acc** is used to quantify the actual performance in predicting a given class as
(4)$$\mathrm{Acc}=\frac{TP+TN}{TN+TP+FP+FN}.$$**Youden’s** The Youden index (or J) denotes the classification threshold for which J is maximal and is defined as [60]
(5)Youden’s=Sn+Sp−1

## 3. Results

This section presents the outcomes of applying ML modeling to the selected datasets. The outcomes of the primary feature selection process utilizing an SVM for each ML model are depicted in Figure 2 for Zou’s ML models and in Figure 3 for Ding’s ML models.

The feature set remains consistent for training the SVM model; however, for the ANN and RF, it was necessary to fine-tune the feature selection. Details of the retained predictors and their relative contributions to the RF model’s performance are illustrated in Figure 4 and Figure 5.

The set of predictors for the ANN model was the same as that for the RF model across both datasets, with the purpose of researching the capabilities and differences in classification for both algorithms. To ascertain the ideal number of features for each algorithm, we identified the point at which the addition of further features to the model yielded marginal improvements in accuracy, as assessed by the Acc metric.

Given the high number of bacteria genera found within the GMC, we considered 18 as a reasonable number of features that can explain most of the classification process and interpret the results; therefore, this number of predictors was used, although RF for Zou’s model can mostly be explained by *Lachnospira* and *Escherichia–Shigella*. The rest of the models for both datasets showed a progressive decrease in performance with the addition of more features. The ANN and SVM models performed with comparable metrics. All the of models were tested on separate data to evaluate the classification performance with unseen cases. Although the ideal would be to test the models with different datasets, it was not possible with this work and will be considered in future research.

### 3.1. Zou’s Dataset

In this dataset, both the support vector machine (SVM) and artificial neural network (ANN) exhibited comparable performance levels, achieving an accuracy of 90% in the test partition as can be seen in Figure 6 and Figure 7.

In contrast, the random forest (RF) model displayed a noticeably lower performance, attaining an accuracy of 80% in the test partition as shown in Figure 8. The RF model’s accuracy ranged from 78% to 83%, and the reported result represents an average of five models, each utilizing five-fold cross-validation. This cross-validation approach involved adjusting the test partitions while maintaining a consistent maximum of trees set to 100.

The generated models using each ML algorithm were compared based on their final accuracy in predicting the target variable against actual observations. The overall average precision of the three models for Zou’s dataset, focusing solely on the test partition, amounted to 86.66%. Notably, the metrics for Youden’s statistics, sensitivity, and specificity for both the SVM and ANN models are presented in Table 1. It is important to remember that, in clinical settings, false negatives are generally considered more critical than false positives, primarily because subjects may not receive the necessary treatment.

Both the SVM and RF algorithms provided insights into feature importance. The top five main features are outlined in Table 2, while the comprehensive feature importance for the SVM model is presented in descending order in Figure 2, and for the RF model in Figure 4. This characteristic proves valuable when constructing models that emphasize the significance of these variables, effectively isolating them from the less impactful bacterial factors.

The SVM model classified 9/10 cases accurately within the test partition. Remember that the test partition comprises data not used for training nor validation. In other words, the developed model tries to classify new events, simulating new clinical cases. The confusion matrix is shown in Figure 6, and in terms of percentages, for the training partition, the accuracy was 91.04%, for the validation 78.94%, and 90% for the test partition. The only error in the prediction was an ASD case classified as NT.

The ANN performed better than the SVM in the training and validation partitions, with 97.01% for training and 82.21% for validation, with the same performance as the SVM in the test partition. The high accuracy in the test partition discards overfitting, as the high accuracy in the training could suggest. Figure 9 shows the architecture for this ANN model.

As in the SVM model, the only misclassification was an ASD case predicted as NT. Most relevant for these two models is that no NT case was misclassified as ASD. This is a remarkable outcome produced with the implementation of an ANN as a potential approximation of the relationships between the human gut composition and ASD symptoms.

### 3.2. Ding’s Dataset

The performance of the SVM for the training partition was 83%, while for the test partition it was 92.3%. The confusion matrix for this model is presented in Figure 10. The relative variable importance for this model is shown in Figure 3. The ANN model reached a performance of 79% for the training partition, 76.92% for the validation, and 92.3% in the test partition. The architecture of the ANN is shown in Figure 11 and the metrics for the ML models are presented in Table 3.

The five most significant variables used for the prediction power of this model were *Lachnospira*, *Bacteroides*, *Lachnoclostridium*, *Blautia*, and *Subdoligranulum*. The RF model reached a constant 92.3% in the test partition using five models with five-fold cross-validation.

The ANN model presents a tendency for the misclassification of ASD cases and that tendency is carried on from the validation partition to the test partition. A possible explanation is that the ASD and NT compositions for those misclassified cases were probably not different enough to be detectable even for ML algorithms, suggesting that those cases could not be associated with GMC and may fall within the approximately 30% of genetically explainable cases. This hypothesis cannot be confirmed purely using ML models and must be confirmed with clinical assessment.

The variable importance chart for the RF model is shown in Figure 5, and its confusion matrix presented for the training, validation, and test partitions is in Figure 12. The average performance for the three models remains at 92.3%, with only one case misclassified for 13 cases not shown to the models before. The test partitions are different among the three models. The only error in the classification in the three models was an NT case classified as ASD.

Table 4 lists a comparison of the five main predictors after fine-tuning the ANN and RF models, with *Ruminococcus torques* and *Anaerobutyricum* being present in the three classifiers. The results show that the SVM and ANN had a similar performance, while RF had a slightly lower accuracy. However, the ANN tends to misclassify some ASD cases as NT, and although this may have a clinical explanation, with the developed tools, such an explanation can only be hypothesized. The RF model’s performance for Zou’s dataset reached an acceptable 75% performance with fewer features, considering that most of the models are related to *Lachnospira*. However, a choice was made to keep all the predictors as future simulations may help to understand the actual contribution of the metabolites produced by those bacteria and their interaction within their environment to modulate the brain function. The main benefit of training and analyzing an ML model based on the GMC of the subjects lies in the elimination of the need to formulate any prior hypotheses about which bacteria are linked to ASD; the results give insights for further research and our efforts to diagnose ASD through ML and the GMC approach complement widely accepted tests. This approach has the potential to expedite the diagnostic process.

## 4. General Discussion

This study’s primary focus was on developing ML models for classifying ASD subjects based on their gut microbiota composition using publicly available datasets. The first dataset originated from the USA, while the second dataset was sourced from China. Two distinct ML models (ANN- and SVM-based) were developed to differentiate between individuals diagnosed with ASD and those without. Notice that these models are not interchangeable nor generalized; they are specific to the GMCs used for training. Additionally, the models are suitable for subjects who closely resemble the sample subjects in terms of age and BMI, and who have not taken antibiotics in the past month [23,24]. The classification capabilities of the proposed models (once confirmed through clinical trials) can streamline the diagnostic process for ASD while maintaining the established standard of care.

The model’s development steps involving data splitting, hyperparameter tuning, model training, and testing were iterated five times to ensure a robust performance and mitigate the impact of variations in splits.

A two-by-two confusion matrix was generated for each dataset fragmentation, providing the counts for true positives, false positives, false negatives, and true negatives [60]. For ASD detection, key metrics such as accuracy, sensitivity, and specificity were calculated based on the values in the confusion matrix and presented in Figure 10, Figure 12 and Figure 13 for the SVM, RF and ANN models respectively. According to the confusion matrix analysis, the varying prevalence of ASD could potentially influence the predictive capabilities of the models, potentially limiting their generalizability.

Previous research has suggested that ML algorithms can effectively reveal intricate relationships between microbiota and neurodevelopmental disorders [61]. Hence, a complementary aim was to identify the bacterial genera predictors that best contribute to early-stage ASD diagnosis and to provide interpretability to the results by understanding how these main predictors might influence brain functionality.

The developed models offer advantages in identifying potential bacteria that can impact the homeostasis of the gut microbiota, often overlooked due to the common practice of disregarding variables that do not reach a pre-established p-value, typically set at 0.05 or lower. Many of the crucial predictors used in our ML models do not exhibit statistically significant differences between ASD and NT subjects. The variations in relative abundances among the majority of the 20 predictors for ASD and NT subjects in both models did not attain the aforementioned statistical significance. This outcome further emphasizes our assertion that attempting to discern the health status of a subject solely based on individual taxa, or even exclusively on those taxa showing statistical differences, is a highly challenging task and may even be unfeasible [62]. The actual health status might arise from intricate interactions among the metabolites produced by a diverse range of bacteria.

Evidence highlights that early ASD treatment can mitigate symptoms, underscoring the importance of accurate and prompt diagnosis for this neurodevelopmental disorder [61]. However, the standalone application of the developed models is not yet practical for clinical use. Further validation is required through clinical practice and testing in other cohorts that share similar characteristics with the training samples used in this study. Real-world decisions must be overseen by medical doctors, making the ML models most valuable for aiding clinicians with diagnosis and treatment decisions [63].

The primary predictor in the RF- and ANN-based models for the first dataset is the genus *Lachnospira*, ranking second in the support vector machine (SVM) model. Following the statistical analysis, some studies employed an RF classifier for data processing [24,64]. However, in Zou’s and Ding’s datasets, the RF algorithm demonstrated the least favorable performance among the ML models.

The family *Lachnospiraceae* does not exhibit statistical differences between ASD and NT subjects in the analysis presented by Zou et al. [23], it is notable that three of the top five predictors in the three models developed in this study for the Zou dataset belong to the family *Lachnospiraceae*. When examining the information published by Zou et al. [23] at the genus level, they report that the statistically distinct genus abundances between ASD and NT children were enriched in ASD subjects: *Bacteroides*, *Prevotella*, *Lachnospiraceae incertae sedis*, and *Megamonas*; meanwhile, these genera were diminished in ASD subjects (enriched in NT): *Clostridium clusters IV and XIVa*, *Eisenbergiella*, *Flavonifractor*, *Escherichia–Shigella*, *Haemophilus*, *Akkermansia*, and *Dialister*.

Within the five main predictors for the SVM and ANN models, only *Bacteroides* ($p=2.4\times 10^{-3}$) is among those reported as statistically different between ASD and NT subjects.

Among the remaining predictors, *Escherichia–Shigella* ($p=2.39\times 10^{-2}$), *Akkermansia* ($p=2.51\times 10^{-2}$), and *Dialister* ($p=3.67\times 10^{-2}$) are statistically different. This brings the total count of statistically different variables to 4 out of the 18 predictors.

The classification accuracy of 90% suggests that the ML algorithms were capable of detecting a complex relationship between those 18 variables and the label ASD in the dataset provided for training. Considering that only 4 of those variables used as predictors in our study presented statistical difference in the analysis made by Zou et al. [23], the rest of the predictors could be considered as relevant and have such complex interactions that they are very hard to find using the classical approach only.

Among the five main important variables in RF was *Escherichia–Shigella*, although *Lachnoclostridium* was not present. An important fact to note is that the ANN/RF and the SVM share the top five predictors, although the order given by each algorithm varies for both classifiers. The RF model had a performance that was 10% lower.

The statistical analysis identifies the two main predictors; however, it may not account for other significant variables that could also influence brain function:The *Blautia* genus, through its metabolites, is capable of mitigating inflammatory and metabolic conditions, as well as having an ability to combat certain microorganisms through antibacterial actions [65] and its strong inflammatory conditions are linked to ASD [66]. The *Eubacterium eligens group* produce Interleukin 10 (IL-10), an anti-inflammatory cytokine that delivers its activity in the epithelial cells [67].*Akkermansia* is considered a novel probiotic candidate that directly influences the gut–brain axis by modulating the permeability of the gut [68]. *Akkermansia* is associated with *Subdoligranulum* and it has been found that when an *Akkermansia* probiotic is consumed, there is also an increase in *Subdoligranulum* [69].*Lachnoclostridium* is a producer of trimethylamine [70], a metabolite that has previously been associated with neurodevelopmental disorders and specifically with the presence of ASD [71,72].Some species of *Feacalibacterium*, such as *Faecalibacterium prausnitzii*, produce short-chain fatty acids (SCFAs) as subproducts of their metabolism. SCFAs contribute to the strength of the intestinal epithelial layer and can reach the brain [73].

In germ-free mice, an increase in the permeability of the blood–brain barrier was observed, and when introducing bacterial strains that produced SCFAs, such as butyrate or acetate/propionate into these germ-free mice, the pattern was reversed and led to a reduction in blood–brain barrier permeability [74]. Other beneficial gut bacteria genuses that produce SCFAs are *Roseburia, Agathobacter, Phascolarctobacterium, and Dialister* [75,76].

The most statistical significant differences at the genus level in descending order reported by Ding et al. [24] are presented in Table 5.

The main predictors obtained in the RF model at the species level generated by Ding et al. are presented in Table 6. However, even with these results, in their study, there is no explanation to justify the main predictors being from a different genus than those found with a bigger statistical difference among the ASD and NT subjects, and no further analysis was presented.

In our RF model corresponding to this dataset, the main five predictors are: *Anaerobutyricum* (ninth in Ding’s model), *Faecalibacterium*, *Clostridium sensu stricto 1 Ruminococcus torques group* (which includes *(Ruminococcus) torques ATCC 27756*, *(Ruminococcus) torques IX–70*, *(Ruminococcus) torques L2–14*, and *(Ruminococcus) torques VIII–23*), and finally *Agathobacter*. Two important points to notice for comparison between our RF model and Ding’s are as follows:There are also species as predictors. As the model trained in the original article used species, we did it this way because it was how the RF model performed the best with fewer predictors. The percentage of the species joined in their genus was low, reinforcing our hypothesis that small changes in low-populated bacteria may have a bigger effect than relatively bigger changes in more populated bacteria.The second point is that the only predictor in common with the RF developed by Ding et al. is *Anaerobutyricum*, which is our main predictor (in the RF model), but in their model is ninth. For the other models, the main five predictors are shown in Table 4.

From the SVM model, which had the best accuracy prediction, the genera *Anaerobutyricum*, *Anaeorstipes*, *Agathobacter*, and *Dorea* are capable of producing SCFAs [77], and in particular, *Anaerostipes and Agathobacter* are mostly butyrate productors [78,79], possibly influencing the presence of ASD by the previously described effects of SCFAs.

As shown for the two datasets, most of the predictors for the trained models are involved in the production of metabolites that are related to inflammation, the permeability of the gut, and SCFAs production.

The ratio of Bacillota/Bacteroidota between ASD and NT subjects was different in the two studies. While Zou et al. found a ratio of Bacteroidota/Bacillota of 0.74 in ASD subjects and 0.31 in the NT group, Ding et al. found no difference (*p* = 0.130). Choosing these two datasets, with their different geographic sampling and results, allowed a better testing scenario for the utility of the application of the ML algorithms to approach this topic.

### 4.1. Limitation of the Study

Similar to many medical studies, this study had a limited sample size that necessitates validation through larger sample sizes in subsequent longitudinal studies and clinical trials. The application of classical ML algorithms lacks the capability to determine causality. Therefore, while the outcomes do not establish a direct causal relationship between the observed disparities in gut microbiota composition and ASD, they have contributed to the formation of a new hypothesis. Furthermore, since the datasets were obtained from public repositories, we did not have control over the original sampling procedures, making it impossible to detect any errors introduced during the sampling process. Future research endeavors should consider incorporating metabolomics data in place of amplicon-based approaches. This shift would enable analyses to be conducted at the species level with enhanced accuracy, thereby improving the outcomes and enabling more precise conclusions.

### 4.2. Future Work

Future work based on this paper can have several branches:Validation of the trained models with other cohorts and their use to classify ASD subjects purely from their microbiota composition.Testing the hypothesis that variables that are not statistically significant may influence complex diseases using other GMC datasets.Research about the intervals of metabolite concentration that are tolerated by the human body before reaching a dysbiosis status.Causal ML can be employed when the prerequisites for such techniques are satisfied, serving as a means to formalize the data-generation process through a structural causal model.

## 5. Conclusions

The presented approach prioritizes the early identification of ASD in children, aiming to enhance their quality of life by enabling swift diagnosis and timely treatment. The effectiveness of the trained models in classifying ASD cases is evaluated through performance metrics. It is evident that the utilization of ML models to detect ASD based on gut microbiota composition yields promising results. Moreover, the performance of these ML models can be enhanced by expanding the dataset to include a larger number of training samples. The results obtained from the assessed classifiers and the relative importance of features suggest that significant changes in the concentration of bacteria with higher prevalence within the gut might be better tolerated without triggering ASD symptoms. Among the various ML tools evaluated, the ANN and SVM emerged as the most effective approaches for establishing a connection between human GMC and the presence of ASD in the subjects under scrutiny. Furthermore, marginal fluctuations in the proportions of less abundant bacteria could potentially contribute to the manifestation of ASD symptoms. Many of the predictors employed for model training deviate from the statistically significant features typically identified. Nonetheless, the metabolites generated by these bacteria, even if not deemed statistically distinct, play a role in inflammatory processes and the production of SCFAs, both of which have been associated with ASD. Addressing this intricate issue may necessitate an interdisciplinary team approach, involving experts in artificial intelligence, particularly ML, to leverage the insights provided by the models. This collaborative effort can help reconcile the conflicting outcomes reported across various research studies concerning the relationship between ASD and GMC.

## Figures and Tables

**Figure 1 biomedicines-11-02633-f001:**
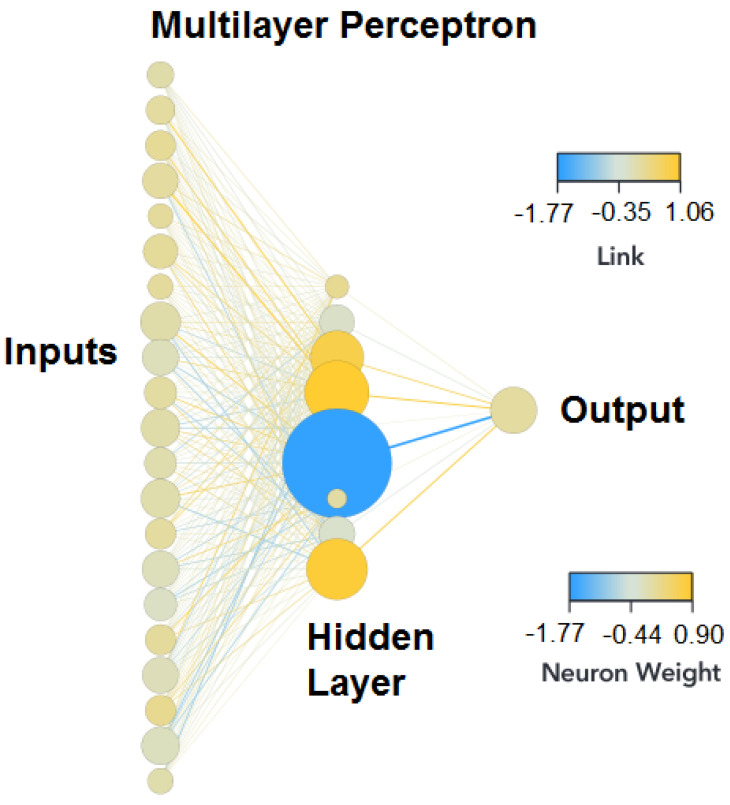
General multilayer perceptron. The visual scale for links and bias goes from blue to yellow, blue to negative values, and yellow to positive ones.

**Figure 2 biomedicines-11-02633-f002:**
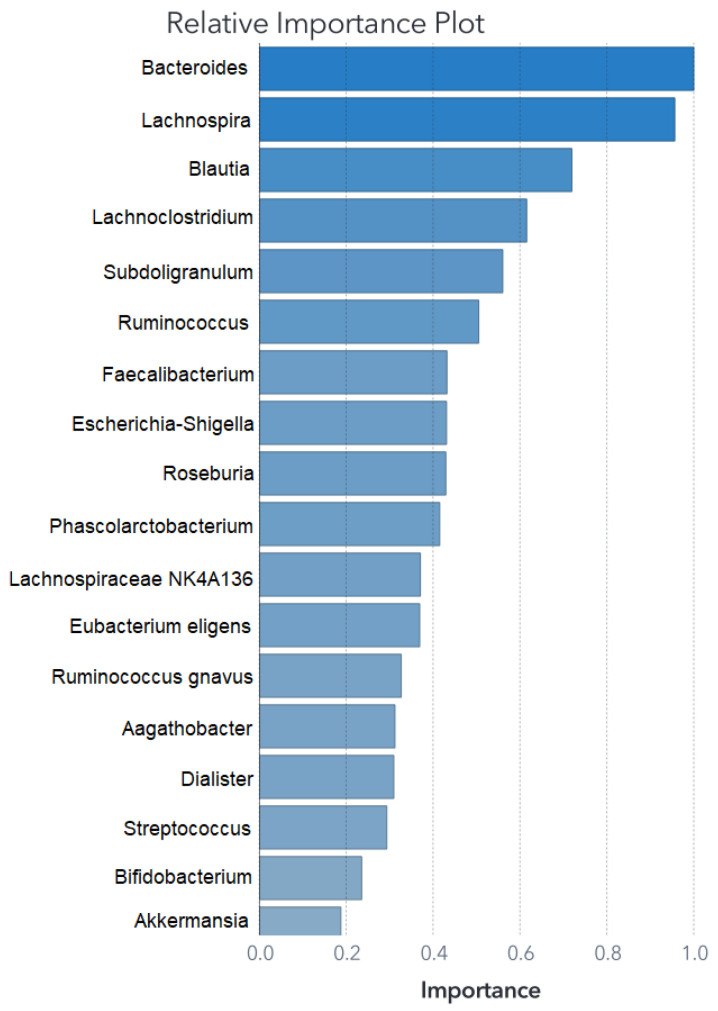
Relative importance for Zou’s SVM model. The first-split log worth is used to rank the variables when applied to the scored training data.

**Figure 3 biomedicines-11-02633-f003:**
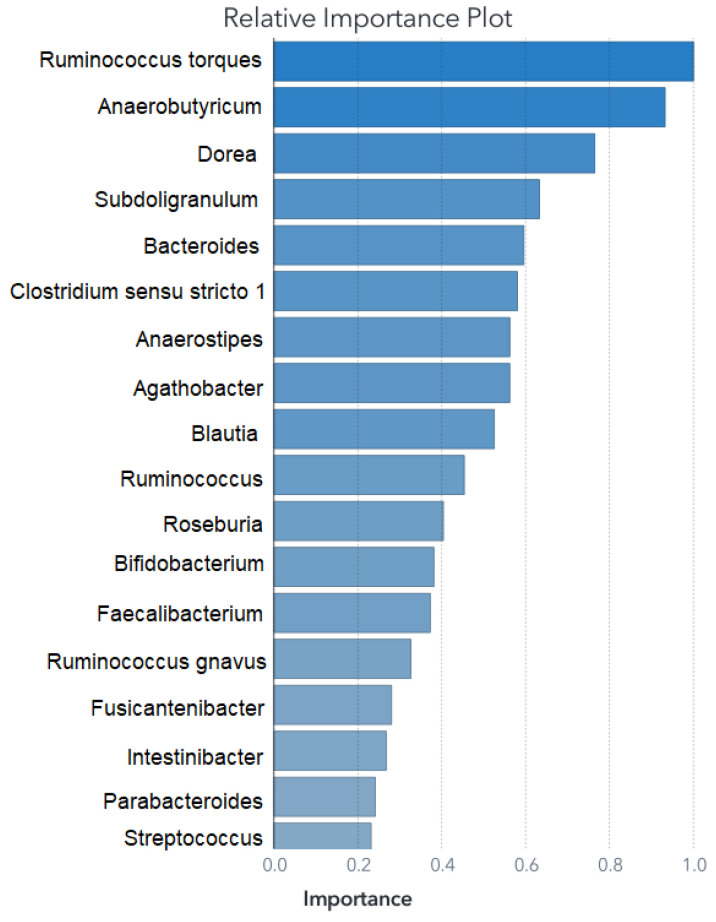
Relative importance for Ding’s SVM model DING. The first-split log worth is used to rank the variables when applied to the scored training data.

**Figure 4 biomedicines-11-02633-f004:**
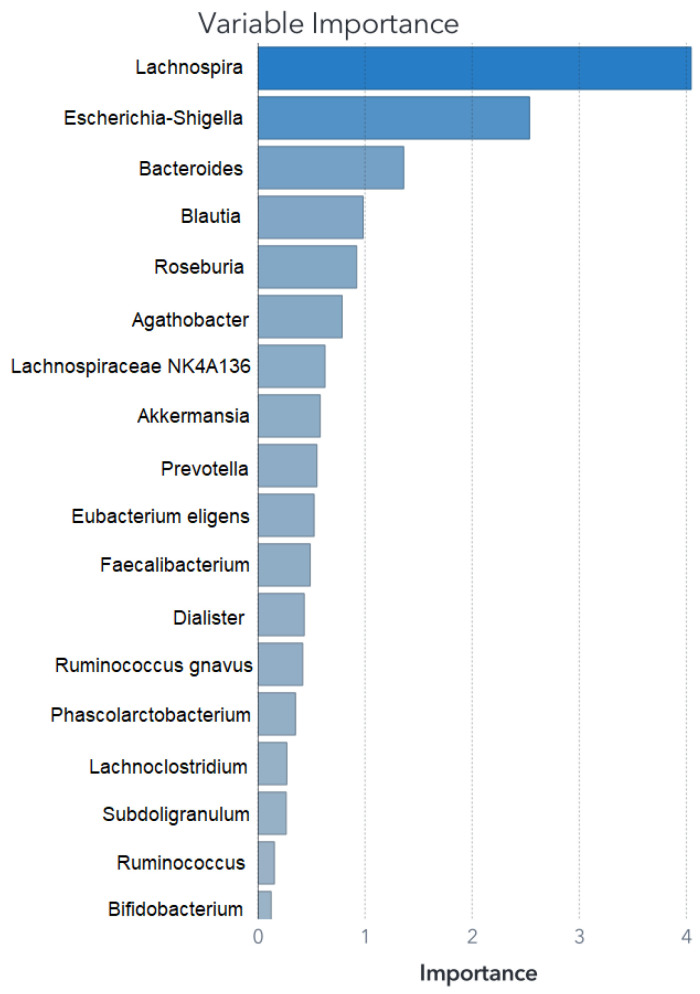
Relative importance for Zou’s RF model. While the RF model does have common variables with the SVM model, the significance attributed to *Lachnospira* stands out significantly compared to the other variables.

**Figure 5 biomedicines-11-02633-f005:**
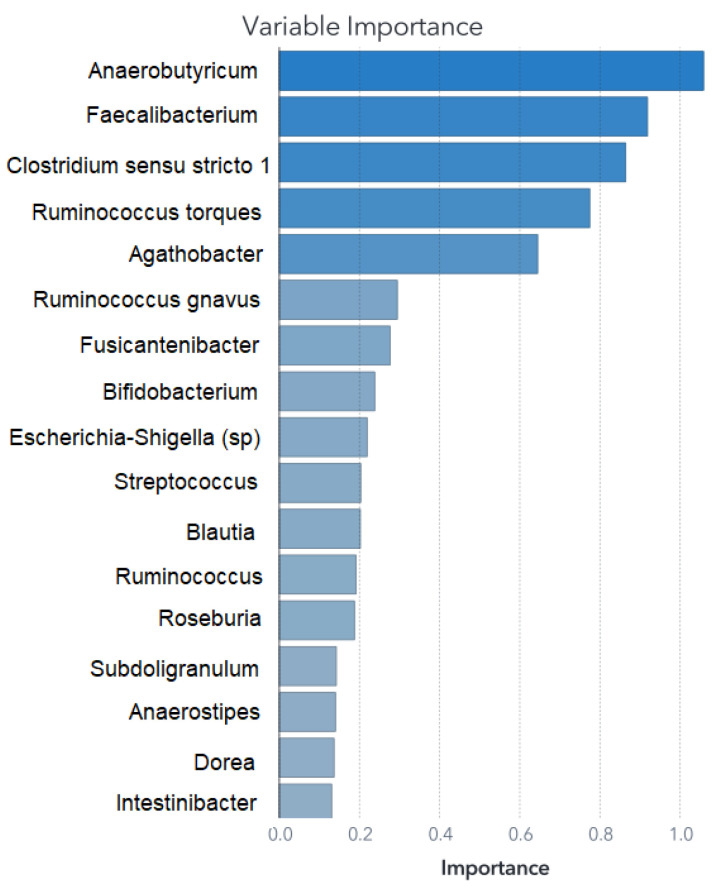
Relative importance for Ding’s RF dataset. Unlike the RF model from Zou’s dataset, in this case, there is not a single dominant predictor; rather, the model’s classification capabilities are mostly attributed to the top five main predictors.

**Figure 6 biomedicines-11-02633-f006:**
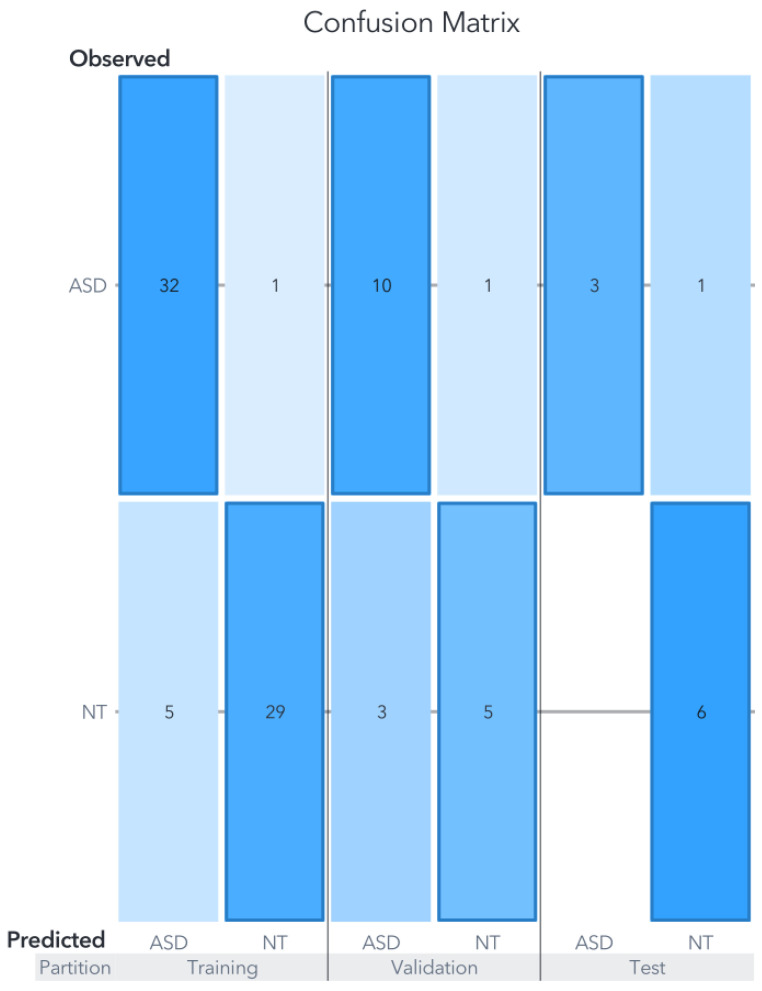
Confusion matrix for Zou’s SVM model. The results of evaluations carried out on the models using data not previously utilized for training or validation are displayed in the confusion matrix labeled under the ”test” category.

**Figure 7 biomedicines-11-02633-f007:**
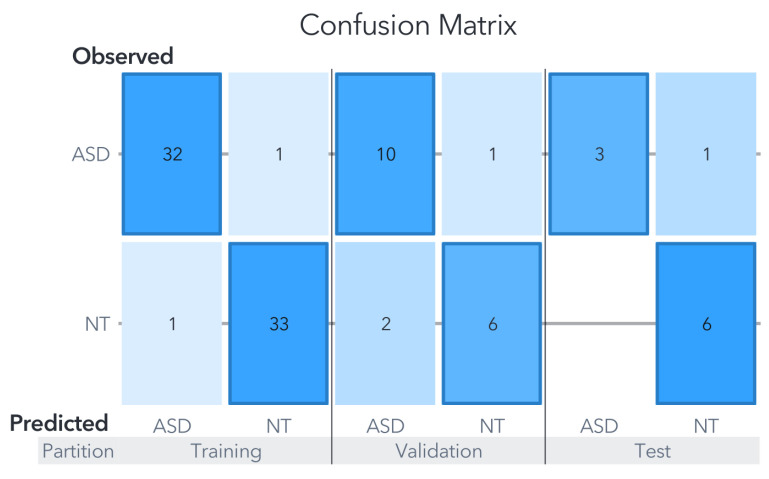
Confusion matrix for Zou’s ANN Model. The ANN model exhibits an intermediate level of accuracy between the RF and SVM models in the training and validation partitions. Nevertheless, it achieves a 90% accuracy in the test partition, surpassing the RF model and matching the performance of the SVM model.

**Figure 8 biomedicines-11-02633-f008:**
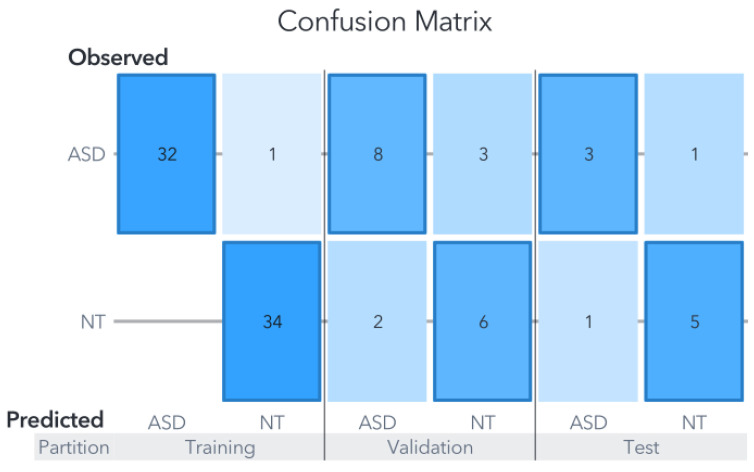
Confusion matrix for Zou’s RF Model. Despite the RF model’s superior performance in the training set compared to the SVM and ANN models, its performance in the test partition was comparatively lower, as it was the only model that misclassified an NT subject.

**Figure 9 biomedicines-11-02633-f009:**
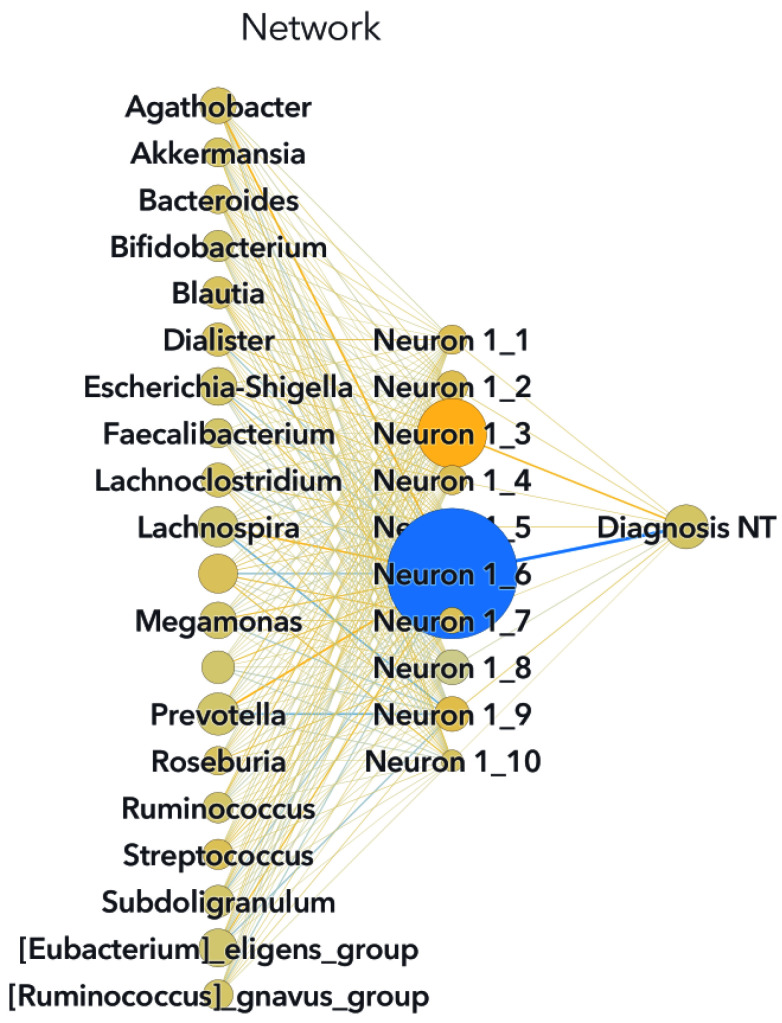
ANN architecture Zou’s dataset.

**Figure 10 biomedicines-11-02633-f010:**
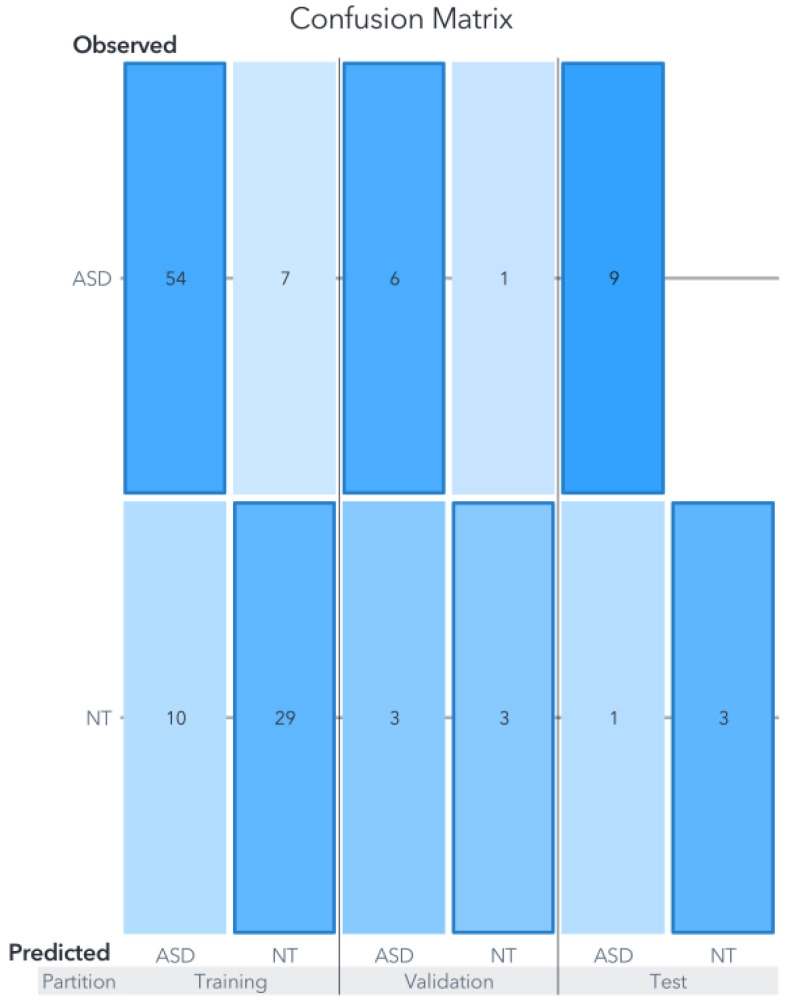
Confusion matrix SVM model Ding. The classification performance achieved in the test partition is better than that in the training and validation partitions, suggesting that overfitting can be discarded.

**Figure 11 biomedicines-11-02633-f011:**
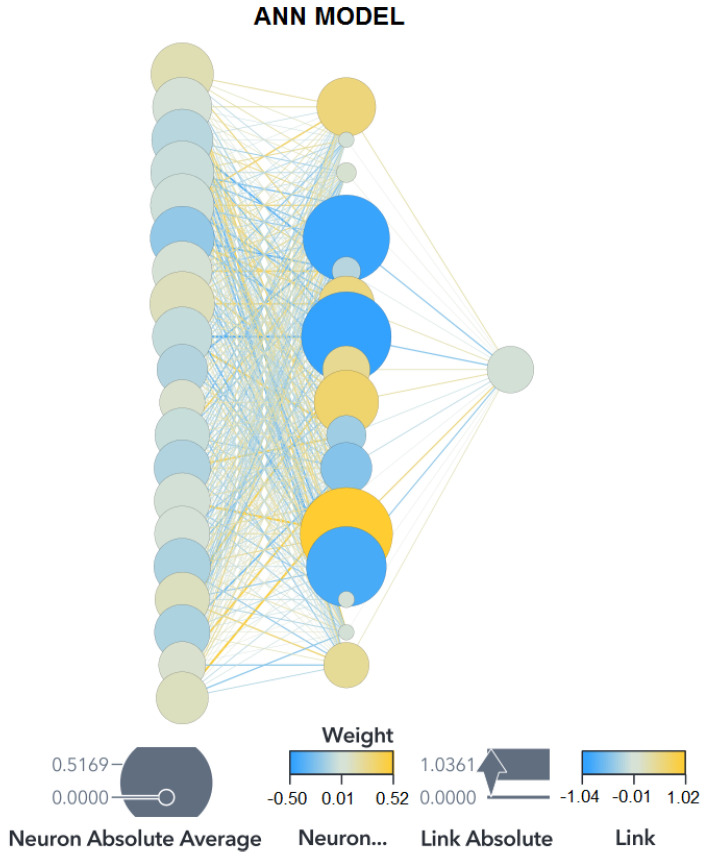
Architecture for ANN model Ding. The weight scale allows us to detect the approximate value of each connection among the neurons, although the actual value is not reported by the software used.

**Figure 12 biomedicines-11-02633-f012:**
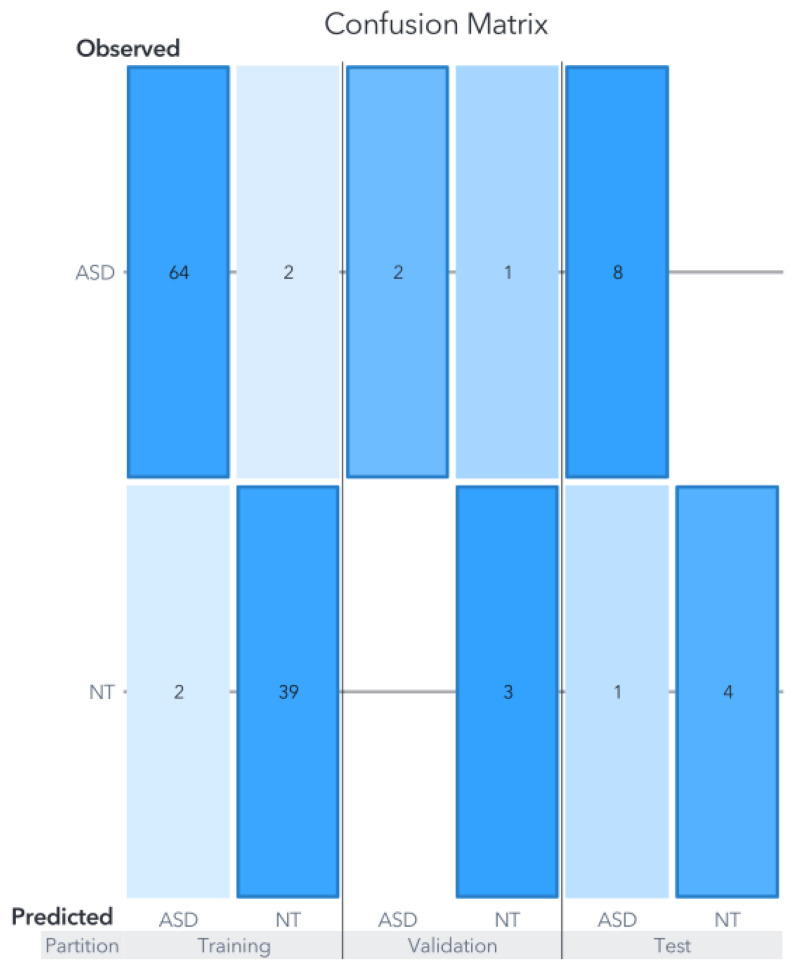
Confusion matrix of the RF model for Ding’s dataset. Different from the ANN model, one misclassification in the partition test cannot be considered as a tendency of the model because such a tendency is not shown in either the validation or the training partitions.

**Figure 13 biomedicines-11-02633-f013:**
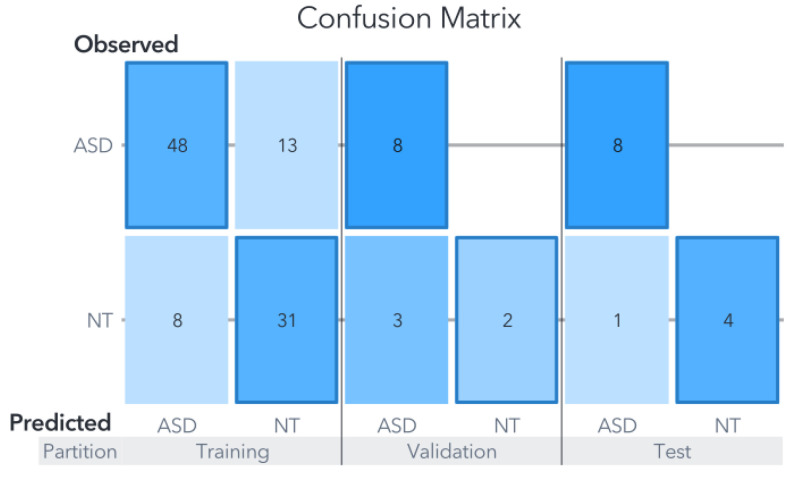
Confusion matrix of the ANN model for Ding’s dataset. Despite achieving an overall accuracy of 92.3%, the model exhibits a slight tendency to classify certain ASD subjects as NT, which is evident in both the validation and test partitions.

**Table 1 biomedicines-11-02633-t001:** SVM and ANN performance indicators in the test partition for the models for Zou’s dataset.

Model	Youden’s	Sensitivity	Specificity	Acc
SVM	0.86	1.0	0.86	0.90
RF	0.58	0.75	0.83	0.80
ANN	0.86	1.0	0.86	0.90

**Table 2 biomedicines-11-02633-t002:** Main predictors for each of the ML classifier models in descending importance for the classification process for Zou’s dataset.

SVM	ANN	RF
Bacteroides	Lachnospira	Lachnospira
Lachnospira	Bacteroides	Escherichia–Shigella
Blautia	Lachnoclostridium	Bacteroides
Lachnoclostridium	Blautia	Blautia
Subdoligranulum	Subdoligranulum	Roseburia

**Table 3 biomedicines-11-02633-t003:** Performance metrics for the models developed with Ding’s dataset.

Model	Youden’s	Sensitivity	Specificity	Acc
RF	0.89	0.89	1.0	0.92
SVM	0.90	0.90	1.00	0.92
ANN	0.89	0.89	1.0	0.92

**Table 4 biomedicines-11-02633-t004:** Main predictors for each of the ML classifiers in descending importance for Ding’s dataset.

SVM	ANN	RF
Ruminococcus torques	Anaerobutyricum	Anaerobutyricum
Anaerobutyricum	Bacteroides	Faecalibacterium
Dorea	Ruminococcus torques	Clostridium sensu stricto
Subdoligranulum	Dorea	Ruminococcus torques
Bacteroides	Subdoligranulum	Agathobacter

**Table 5 biomedicines-11-02633-t005:** Variables that show statistical difference according to Ding et al. [24].

Genus	Relative Presences	*p*-Value
*Bacteroides*	16.89% ASD vs. 22.69% NT	*p* = 0.011
*Faecalibacterium*	7.87% ASD vs. 10.20% NT	*p* = 0.038
unidentified *Lachnospiraceae*	6.89% ASD vs. 4.58% NT	*p* < 0.001
unidentified *Clostridiales*	1.50% ASD vs. 0.77% NT	*p* < 0.001
*Dorea*	1.27% ASD vs. 0.76% NT	*p* < 0.001
unidentified *Erysipelotrichaceae*	1.04% ASD vs. 0.66% NT	*p* < 0.016
*Collinsella*	0.97% ASD vs. 0.36% NT	*p* < 0.004
*Lachnoclostridium*	0.80% ASD vs. 0.54% NT	*p* < 0.001
*Parasutterella*	0.10% ASD vs. 0.31% NT	*p* = 0.001
*Paraprevotella*	0.01% ASD vs. 0.11% NT	*p* = 0.034

**Table 6 biomedicines-11-02633-t006:** Predictors from the species level RF model reported by Ding et al. [24].

Species	Genus	Genus Predictor in Our Models
*Bifidobacterium Bifidum*	*Bifidobacterium*	Yes
*Coprococcus eutactus*	*Coprococcus*	No
*Streptococcus mutans*	*Streptococcus*	Yes
*Anaerostipes caccae*	*Anaerostipes*	Yes
*Anaerobutyricum halli*	*Anaerobutyricum*	No
*Collinsella aerofaciens*	*Collinsella*	No
*Eubacterium coprostanoligenes*	*Eubacterium*	Yes
*Tyzzerella sp Marseilee P30+3*	*Tyzzerella*	No
*Roseburia intestinalis*	*Roseburia*	Yes
*Clostridium sp cTpY 17*	*Clostridium*	Yes

## Data Availability

The sequence data for the microbiome composition were downloaded from the GeneBank Sequence Read Archive with the accession number PRJNA578223 and are from the research conducted by Zou et al. [23]. These data can be found at https://www.ncbi.nlm.nih.gov/bioproject/PRJNA578223/, accessed on 1 March 2022. The second dataset was downloaded from the NCBI Biosample Database with the accession number PRJNA589343, and is from the research conducted by Ding et al. [24]. These data can be found at https://www.ncbi.nlm.nih.gov/bioproject/PRJNA589343/, accessed on 1 March 2022.

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
