# Peer review of "Machine Learning Algorithms Applied to Predict Autism Spectrum Disorder Based on Gut Microbiome Composition"

_biomedicines, 2023, doi:10.3390/biomedicines11102633_

Round 1
Reviewer 1 Report
Authors are not experts in biology and also extensive corrections in taxonomical nomenclature and associations are recommended. Furthermore, within the field of informatics, there are parts of the manuscript where more clear, concise and higher level scientific language is required. Results are insufficient for publishing and the discussion is inappropriate.

Extensive editing of English language required.
Author Response
Authors are not experts in biology and also extensive corrections in taxonomical nomenclature and associations are recommended. Furthermore, within the field of informatics, there are parts of the manuscript where more clear, concise and higher level scientific language is required. Results are insufficient for publishing and the discussion is inappropriate.
Response: Thanks a lot for taking the time to review our manuscript. We have carefully read all of your comments and answered them. We hope all of them have been addressed adequately.
Please find attached to this answer the document with the corrections highlighted.
Regards
The authors
- It is not true, that previous studies focused on the most abundant genera.
Response: Thanks for pointing out this fact, but according to our research we found several articles supporting the point. In the study from Pulikkan et al. (doi:10.1007/s00248-018-1176-2), there is a subsection in "Results" entitled "Taxonomic Comparison of Mayor Taxa in ASD and Healthy Children," and the further analysis about the differences in gut microbiota composition in ASD and NT subjects is based on that. Afterward, in that same subsection, it is written, "A comparison of major genera between ASD and healthy groups revealed a significantly higher abundance of…." and proceeds to describe those variables and their statistical significance. Most of the analysis in that section is made in those most abundant families and genera, while the differences for low abundant families are just mentioned in 3 rows and are not further analyzed.
In the study from Zou et al. (DOI: 10.1002/aur.2358 ), one of the datasets used in our work, a clear preference by the most abundant phyla, families, and genera is taken for the analysis. "Four major abundant phyla…" "..showed significant variations in abundance between the two groups" about the families, it is written, "six major families had significantly different abundances between the ASD and the control" and "At the genus level, a total of 12 major genera had significant difference…." "At the species level, we found 11 abundant species with significantly different abundances…." Similar to Pulikkan et al., there is little attention to the less abundant taxa, and just a few rows are dedicated to their analysis.
Tomova et al. (DOI: 10.1016/j.physbeh.2014.10.033) write: "Although over 50 bacterial phyla have been described in the human gut microbiota, we focused our study on the two most dominant" and this is also considered when the analysis of genera is made.
We sincerely believe that the presentation of these three references has effectively conveyed our point. However, thanks to your review and given the fact that we cannot be 100% sure that all the previous studies are mainly focused on the most abundant phyla, families, and genera, we changed the manuscript from "considering that previous results have focused in the most abundant genera…" to "considering that several previous studies have focused in the most abundant genera, in particular those used as datasets in this work, …"
2. What is ment the traditional effect on ASD evolution?
Response: The traditional effect of Gut Microbiome in the ASD evolution is that “imbalance in the composition of the microbiota can result in disturbed host-microbiota homeostasis which can lead to ASD” as stated in the paper entitle Gut microbiota and Autism Spectrum Disorder: From pathogenesis to potential therapeutic perspectives (https://doi.org/10.1016/j.jtcme.2022.03.001).
We have rewritten the phrase in the manuscript to clarify it. Now it is stated as “wherein an imbalance in the composition of the microbiota can lead to disrupted host-microbiota homeostasis”.
3. This is insufficient outcome of the study stating only the importance of less abundant genera analysis.
Response: The importance of less abundant genera analysis is not the only outcome of the study. The creation of accurate models after passing the steps of training, debugging, testing, and implementation of the models for diagnosis of ASD based on the GMC are, by themselves, relevant outcomes that may help improve accuracy in the diagnosis and may help with clustering ASD cases. As a significant example, if there is an ASD subject that is classified as NT by a majority vote of the models, it could be an insight that a person's ASD status may not be linked to GM dysbiosis. Given that the machine learning models “requires fewer assumptions about the underlying relationships between the data elements” (Bennett 2018), it can also help to direct research in approaches not previously considered. Accurate models have several applications and are valuable specially for diagnosing and classifying complex diseases. As a reference for addressing the importance of ML in biomedicine, the article How Machine Learning will Transform Biomedicine found in 10.1016/j.cell.2020.03.022 can help to support this claim.
The importance of less abundant genera is just one of the study's outcomes, but if it is proven in clinical trials, it could be the most important. This fact was realized after developing the first model and observing that the predictors included abundant genera and not previously considered lesser abundant genera. This led us to test that hypothesis in a second dataset (Gut microbiota changes in patients with autism spectrum disorders DOI: 10.1016/j.jpsychires.2020.06.032) with similar results.
4. I am missing the interconnection between the detected bacterial genera and different diets. Please mention in the abstract what are the outcomes of the study considering diets.
Response: We prefer to remove any claim associated with bacterial genera and their relationship with the different diets, we understand the current evidence could be not enough. Until now we just analyzed two papers associated (Gut microbiota changes in patients with autism spectrum disorders DOI: 10.1016/j.jpsychires.2020.06.032 and Changes in the Gut Microbiota of Children with Autism Spectrum Disorder DOI: 10.1186/s13099-020-0346-1) with this idea. Even though our conclusion is correct we can not sustain this argument based on previously reported studies, it could be an interesting idea to explore in a future work.
5. reformulate
Response: It has been changed from “Those results give some clues about their relationship” to “The outcomes of the aforementioned studies have yielded valuable insights into the relationship between ASD and GMC”
6. replace with "excluding"
Response: Changed “leaving outside” with “excluding”
7. This part is not related to the topic, please be more specific and focused on ASD.
Response: The whole paragraph “Type 1 Diabetes (T1D) in rat models has already been associated with the decline of Firmicutes and the abundance of Bacteroidetes in microbiome, according to the study presented in (Giongo2011)” has been removed as per your kindly suggestion.
8. replace the taxonomical nomenclature with the uptodate one; Firmicutes has beem repleaced by Bacillota
Response: Thank you for your observation. We appreciate your feedback and have taken steps to ensure accuracy in our content. In line with the updated taxonomical understanding, we have replaced Firmicutes for Bacillota
(https://doi.org/10.1099/ijsem.0.005056)
9. replace with Bacteroidota
Response: Thank you for your observation. We appreciate your feedback and have taken steps to ensure accuracy in our content. In line with the updated taxonomical understanding, we have replaced Bacteroidetes with Bacteroidota (https://doi.org/10.1099/ijsem.0.005056)
10. Which study?
Response: Corrected from “The study concluded…” to “In 2012, Yatsunenko et al. concluded…”
11.its
Response: Corrected “their” to “its”
12. Please match the stating that you wish to find causation instead of correlation, with the obtained results and include them in the abstract. How can you prove that you detected the causation?
Response: We deeply regret any confusion caused. While our research goal is centered on identifying causal relationships using machine learning, this study does not yield such findings. Our ongoing work is dedicated to achieving this goal of establishing causation, and we firmly believe that this present article serves as a foundational step, guiding our future research endeavors towards a more comprehensive understanding of causation. The paragraph 'Considering that machine learning methods could meet human experts in diagnosing some diseases [15,16], there are good odds of being capable of exceeding our capability in finding causation variables between ASD and gut microbiome evolution,' suggests that given machine learning ability to match the diagnostic capabilities of medical experts, it's plausible that machine learning could be better at identifying causal variables than humans as well. This is particularly relevant in the context of ASD and gut microbiome evolution, where humans have struggled to achieve this goal, possibly indicating that such causation might not exist at all. It is crucial to emphasize that we do not claim to have established causation within our results. In fact, the existence of such causation remains uncertain. Our research is driven by the quest to uncover potential causal relationships, motivated by our recognition of this uncertainty.
13. Add citation
Response: Added the references Diagnostic and statistical manual of mental disorders DSM-5 (https://doi.org/10.1176/appi.books.9780890425596), Autism Diagnostic Interview Revised (https://doi.org/10.1007/978-0-387-79948-3_1519), The Autism Diagnostic Observation Schedule--Toddler Module (https://doi.org/10.1007/s10803-009-0746-z) and Childhood Autism Rating Scale (https://doi.org/10.1177/0734282911400873).
14. reformulate the sentence
Response: we reformulate the sentence from “the root causes are not clear yet” to “the root causes remain unclear” and merge with the previous sentence to be “The rate of ASD among newborns rising since the mid-20th century is a known fact, but the root causes remain unclear”
15. approximately
Response:Changed “Approximate the” to “Approximately”
16. reformulate the sentence
Response: Changed from “In the middle of those approaches, a growing research field relates the gut microbiome composition with ASD manifestation, with several studies supporting this approach” to “Among these lines of investigation, a burgeoning research area centers around the correlation between the composition of the gut microbiome and the manifestation of ASD, an approach substantiated by several supportive studies”
17. Be more specific about cohort of individuals also in other studies mentioned. What was their age, sex...?
Response: Added the information that the origin study provided about the samples: “The first dataset consist of 96 fecal samples, 48 from ASD children (from 2 to 7 years old, average 5, average BMI=17.4, 10 females and 38 males) and 48 from NT children (all at 48 months, no allergies, 24 females and 24 males, average BMI=16.3). The high-quality 16S rRNA sequences were obtained from the 96 samples following high-throughput DNA sequencing. The total gut microbiota was determined through the phylogenetic and taxonomic assessments of the 16S rRNA V3-V4 regions.”
18. replace with neurotypical subjects
Response: Changed “Typically Developing Behavior (TD) ” to “Neurotypical (NT)”
19. geographic locality?
Response: We believe that the phrase “insite location” reflects better how the samples were obtained, hence we decided to keep this phrase as stated in the original paper by Fouquier et al.
20. Reformulate the sentence.
Response: The sentence was corrected from “In 2021, a work conducted by Fouquier et al.” to “In 2021 Fouquier et al.” and “They took their samples in Arizona and Colorado, both within the USA, which leads to an intuition that studying microbiome compositions and their influence on ASD from samples taken in several countries must be an even more complex task that requires modern approaches.” was changed to “Analyzing microbiome compositions and their impact on ASD using samples collected across multiple countries poses an even greater challenge, demanding the utilization of modern advanced methodologies (such as machine learning tools)”
21. its
Response: In this case we are using “their” because we are referring to the plural “compositions”, as we are talking about the GMC of different places. Using “its” would require the change to singular and would change the meaning of the sentence.
22. it
Response: “a variable was considered if they had a value of at least 0.01” was changed to “a variable was considered if it had a value of at least 0.01”
23. Clostridium has higher relative abundance? Clostridium cannot be higher.
Response: Replaced “Clostridium is higher” with “Clostridium has higher relative abundance”
24. Names of the taxa should be capitalised. Wrong taxonomical name.
Response: Corrected “firmicutes/bacteroidetes” with Bacillota/Bacteroidota
25. Do you mean big difference in the ratio?
Response: Changed “a big rate is found in people diagnosed with ASD” to “a big Bacillota/Bacteroidota ratio is found in people diagnosed with ASD.”
26. reformulate
Response: Replaced “conflicting results happen” with “conflicting outcomes arise”
27. The word should not be written in cursive.
Response: Changed “and” to “and”
28. It is not true,
Response: The following is a transcription of a paragraph from Banerjee et al. in their article Probability, clinical decision making and hypothesis testing criticizing the interpretation of p values : "According to convention, the results of p < .05 are said to be statistically significant, and those with p > .05 are said to be statistically nonsignificant. These expressions are taken so seriously by most that it is almost considered ‘unscientific’ to believe in a nonsignificant result or not to believe in a ‘significant’ result. It is taken for granted that a ‘significant’ difference is a true difference and that a ‘nonsignificant’ difference is a chance finding and does not merit further exploration" (doi: 10.4103/0972-6748.57864).
In our manuscript, a similar situation is presented: The authors (DOI: 10.1016/j.jpsychires.2020.06.032 and DOI: 10.1186/s13099-020-0346-1) that made the statistical analysis focused only on those bacteria that presented a statistical difference considering p < .05, and no further analyzed the importance of bacteria that did not reach such value. However, we considered it wise to replace the sentence "involves considering only statistically significant variables to establish causation" with "involves mainly considering significance tests to establish causation and accept or reject hypotheses"
29. abundance
Response: Replaced “relative presence” with “abundance”
30. Obtained results are neither described nor discussed. It is rather the principle of the machine learning tool applied what is not the main point. It is also very confusing wether the article is focused on the comparison of machine learning tools or identification of bacteria and it is not clearly stated.
Response: We have rewritten this section in order to clarify the objective of this paper.
This study's primary focus was on developing ML models for classifying ASD subjects based on their gut microbiota composition using publicly available datasets. The first dataset originated from the USA, while the second dataset was sourced from China.
Two distinct ML models (ANN and SVM based) were developed to differentiate between individuals diagnosed with ASD and those without. Notice that these models are not interchangeable; they are specific to the used GMCs for training. Additionally, the models are suitable for subjects who closely resemble the sample subjects in terms of age, BMI, and who have not taken antibiotics in the past month. (DOI: 10.1002/aur.2358 and DOI: 10.1016/j.jpsychires.2020.06.032).
The classification capabilities of the proposed models (once confirmed through clinical trials) can streamline the diagnostic process for ASD, while maintaining the established standard of care.
The model development steps involving data splitting, hyperparameter tuning, model training, and testing were iterated five times to ensure a robust performance and mitigate the impact of variations in splits.
A two-by-two confusion matrix was generated for each dataset fragmentation, providing the counts for true positives, false positives, false negatives, and true negatives. (DOI Berrar2019) For ASD detection, key metrics such as accuracy, sensitivity, Youden’s index and specificity were calculated based on the values in the confusion matrix. According to the confusion matrix analysis, the varying prevalence of ASD could potentially influence the predictive capabilities of the models, potentially limiting their generalizability.
Previous research has suggested that ML algorithms can effectively reveal intricate relationships between microbiota and neurodevelopmental disorders (https://doi.org/10.1016/j.bbe.2020.01.008). Hence, a complementary aim was to identify the bacterial genera predictors that best contribute to early-stage ASD diagnosis and to provide interpretability to the results by understanding how these main predictors might influence brain functionality.
The developed models offer advantages in identifying potential bacteria that can impact the homeostasis of the gut microbiota, often overlooked due to the common practice of disregarding variables that do not reach a pre-established p-value, typically set at 0.05 or lower. Many of the crucial predictors used in our ML models do not exhibit statistically significant differences between ASD and NT subjects. The variations in relative abundances among the majority of the 20 predictors for ASD and NT subjects in both models did not attain the aforementioned statistical significance. This outcome further emphasizes our assertion that attempting to discern the health status of a subject solely based on individual taxa, or even exclusively on those taxa showing statistical differences, is a highly challenging task and may even be unfeasible (https://doi.org/10.1128/mbio.00434-20). The actual health status might arise from intricate interactions among the metabolites produced by a diverse range of bacteria.
Evidence highlights that early ASD treatment can mitigate symptoms, underscoring the importance of accurate and prompt diagnosis for this neurodevelopmental disorder. (https://doi.org/10.1016/j.bbe.2020.01.008). However, consider that the standalone application of the developed models is not yet practical for clinical use. Further validation is required through clinical practice and testing in other cohorts that share similar characteristics with the training samples used in this study. Real-world decisions must be overseen by medical doctors, making the ML models most valuable in aiding clinicians with diagnosis and treatment decisions (DOI: 10.1016/S2589-7500(21)00041-8).
31. All ML tools applied should be mentioned.
Response: Replaced “(Naive Bayes, kNN, Ensemble Learning, Logistic Regression, etc)” with “(Naive Bayes, k-Nearest Neighborhood, Ensemble Learning, Logistic Regression, Linear Regression and Decision Trees)”.
32. This text should be associated with the picture, not included within the text.
Response: Changed the placement of the description of the figure and removed it from the text.
33. It is not clear, whether it was a dataset of metagnenomes or amplicons, please be clear.
Response: We reformulate the sentence to clear it out “The first database comprised 96 gut metagenomes, 48 from ASD children and 48 from NT children. In this dataset, 38 of the 48 ASD children and 24 of the 48 NT children were males. The genomic DNA was extracted from amplifications of the regions V3 and V4 of the 16S rRNA genes.” to “The first dataset consisted of 96 fecal samples, 48 from ASD children (from 2 to 7 years old, average 5, average BMI=17.4, 10 females and 38 males) and 48 from NT children (all at 48 months, no allergies, 24 females and 24 males, average BMI=16.3). The high-quality 16S rRNA sequences were obtained from the 96 samples following high-throughput DNA sequencing. The total gut microbiota was determined through the phylogenetic and taxonomic assessments of the 16S rRNA V3-V4 regions.”
34. There are probably not more amplifications, but you mean amplification of the 16S rRNA gene. Please reformulate the statement.
Response: We are sorry about the confusion, yes we meant amplification of the 16S rRNA in specific from region V4. The text was changed to make the statement clear: “For identification of the gut microbiota, the V4 region of 16S rRNA genes were amplified from DNA isolated from fecal samples.”
35. What do you mean by that? Bacterial genera?
Response: Thanks you for the observation. We reformulate the paragraph, Alpha-rarefaction analysis was performed to verify a sufficient sequencing depth using qiime2 v2020, and the classification was taken until the genus level.
from: Alpha-rarefaction analysis was performed to verify a sufficient sequencing depth using qiime2 v2020, and the classification was taken until the genus level, and where possible, until species level.
Once the bioinformatics analysis was completed, the genus containing chloroplasts were removed. The absolute…
to: Alpha-rarefaction analysis was performed to verify a sufficient sequencing depth using qiime2 v2020, and the classification was taken until the genus level.
The absolute…
36. Counts cannot be mapped in this case, please correct the statement.
Response: Changed “The absolute counts were mapped into relative abundances” to “The absolute counts were converted into relative abundances”
37. Please use more concise language.
Response: Changed “and fed this way” to “All the samples were normalized on a scale [0, 1] before being considered as inputs to the proposed classifiers.”
38. Figures are not mentioned in the text. It is difficult to follow Please include them.
Response: We have corrected this issue.
39. The results section is too short and not sufficient for publishing. This chapter is absolutely insufficient.
Response: We have made several improvements in the results section and added details. La respuesta es la misma que para el punto 30 que hay que dividir entre la discusión y los resultados:
This study's primary focus was on developing two ML models for classifying ASD subjects based on their gut microbiota composition using publicly available datasets. The aim was to identify the bacterial genera predictors that best contribute to early-stage ASD diagnosis and to provide interpretability to the results by understanding how these main predictors might influence brain functionality. Previous research has suggested that ML algorithms can effectively reveal intricate relationships between microbiota and neurodevelopmental disorders.
Two distinct models were developed to differentiate between individuals diagnosed with ASD and those without, The first dataset originated from the USA, while the second dataset was sourced from China. It's important to note that these models are not interchangeable; they are specific to the GMCs used for training. Additionally, the models are suitable for subjects who closely resemble the sample subjects in terms of age, BMI, and who have not taken antibiotics in the past month.
The classification capabilities of these models (once confirmed through clinical trials) can streamline the diagnostic process for ASD, while maintaining the established standard of care.
The steps involving data splitting, hyperparameter tuning, model training, and testing were iterated five times to ensure a robust model performance and mitigate the impact of chance variations in splits.
A two-by-two confusion matrix was generated for each dataset fragmentation, providing the counts for true positives, false positives, false negatives, and true negatives. For ASD detection, key metrics such as accuracy, sensitivity, and specificity were calculated based on the values in the confusion matrix.
However, it's important to note that the standalone application of the developed models is not yet practical for clinical use. Further validation is required through clinical practice and testing in other cohorts that share similar characteristics with the training samples. Real-world decisions must be overseen by physicians, making the ML models most valuable in aiding clinicians with diagnosis and treatment decisions.
Our models offer advantages in identifying potential bacteria that can impact the homeostasis of the gut microbiota, often overlooked due to the common practice of disregarding variables that do not reach a pre-established p-value, typically set at 0.05 or lower. Many of the crucial predictors used in our ML models do not exhibit statistically significant differences between ASD and NT subjects.
The variations in relative abundances among the majority of the 20 predictors for ASD and NT subjects in both models did not attain statistical significance. This outcome further emphasizes our assertion that attempting to discern the health status of a subject solely based on individual taxa, or even exclusively on those taxa showing statistical differences, is a highly challenging task and may even be unfeasible. The actual health status might arise from intricate interactions among the metabolites produced by a diverse range of bacteria.
The varying prevalence of ASD could potentially influence the predictive capabilities of the models, potentially limiting their generalizability.
Evidence highlights that early ASD treatment can mitigate symptoms, underscoring the importance of accurate and prompt diagnosis for this neurodevelopmental disorder.. (https://doi.org/10.1016/j.bbe.2020.01.008)
40. These results are presented in Table 2 as Zou´s dataset.
Response: We believe that there is a misunderstanding in this observation, the table 2 is to show the readers the relationship between main predictors that were found in this work. We have rewritten the results hoping to clarify this idea.
41. The discussion is inappropriate and focuses mostly on bacteria identified in the original study and their role. The results should be discussed with relevant publications using ML algorithms for ASD detection, classification.
Response: We have improved the discussion focusing on the results achieved by machine learning and the performance of the different machine learning algorithms that were tested for this work, even though we decided to keep the idea of the bacteria role we believe this could be helpful for other researchers. The changes are highlighted in the attached manuscript.
42. This taxon cannot be considered the same as Lachnospira or Lachnoclostridium, although it belongs to the same family.
Response: We have made the correction.
43. use relevant citations
Response: Thanks a lot for pointing this out. We made some changes to the manuscript, and the relevant citations are mentioned next.
The way the text was written could lead to confusion, so we changed it from: “The statistical analysis finds the two main predictors, however fails to find other important variables that may also have an impact influencing the brain function.
Blautia genus, through its metabolites, is capable of mitigating inflammatory and
metabolic conditions, as well as its ability to combat certain microorganisms through
antibacterial actions [53] and strong inflammatory conditions are linked to ASD [54].
Eubacterium eligens group produce Interleukin 10 (IL-10), an anti-inflammatory cytokine that delivers its activity in the epithelial cells [55].
Akkermansia is considered a novel probiotic candidate that directly influences the gut-brain axis by modulating the permeability of the gut [56]. Akkermansia is associated with Subdoligranulum and it has been found that when Akkermansia probiotic is consumed, there is also an increase in Subdoligranulum [57].
Lachnoclostridium is a producer of trimethylamine [58], a metabolite that previously has been associated with neurodevelopmental disorders and in specific with the presence of ASD [59,60].” to
The statistical analysis finds the two main predictors. However, fails to find other important variables that may also have an impact on brain function:
- Blautia genus, through its metabolites, is capable of mitigating inflammatory and metabolic conditions, as well as its ability to combat certain microorganisms through antibacterial actions [53] and strong inflammatory conditions are linked to ASD [54].
- Eubacterium eligens group produces Interleukin 10 (IL-10), an anti-inflammatory cytokine that delivers its activity in the epithelial cells [55].
- Akkermansia is considered a novel probiotic candidate that directly influences the gut-brain axis by modulating the permeability of the gut [56]. Akkermansia is associated with Subdoligranulum and it has been found that when Akkermansia probiotic is consumed, there is also an increase in Subdoligranulum [57].
- Lachnoclostridium is a producer of trimethylamine [58], a metabolite that previously has been associated with neurodevelopmental disorders and specific with the presence of ASD [59,60].
We are confident that, with these corrections in place, we may claim that the supplementary information regarding other significant variables that might influence brain function can be found below. Furthermore, all of these points are appropriately referenced.

Reviewer 2 Report
Interesting study-
I have some minor suggestions with a pure academic spirit:
1. The abstract must better summarize the sections (it directly starts with the proposal..)
2. “1.1. ASD and GM “ terrible heading. Please avoid acronyms in the titles of the pars.
3. Please describe the figures in details in the body of the ms
4. Insert the limitations in the discussion
Author Response
Comments and Suggestions for Authors
Interesting study-
Response: Thanks a lot for taking the time to review our manuscript. We have carefully read all of your comments and answered them. We hope all of them have been addressed adequately.
Please find attached to this answer the document with the corrections highlighted.
Regards
The authors
Reviewer 2:
I have some minor suggestions with a pure academic spirit:
- The abstract must better summarize the sections (it directly starts with the proposal..)
Response: According to the comments of reviewers the abstract was corrected and we highlight the changes made on it on the attached new version of the document.
- “1.1. ASD and GM “ terrible heading. Please avoid acronyms in the titles of the pars.
Response: The heading was changed to Autism Spectrum Disorder and Gut Microbiota Composition
- Please describe the figures in details in the body of the ms
Response: We have included the figures in the main body of the manuscript and describe in detail the meaning of them.
- Insert the limitations in the discussion
Response: The subsection “Limitations of the study” was included in the discussion section and reproduced next: “Similar to many medical studies, this study had a limited sample size that necessitates validation through larger sample sizes in subsequent longitudinal studies and clinical trials. The application of classical ML algorithms cannot determine causality. Therefore, while the outcomes do not establish a direct causal relationship between the observed disparities in gut microbiota composition and ASD, they have formed a new hypothesis. Furthermore, since the datasets were obtained from public repositories, we did not have control over the original sampling procedures, making it impossible to detect any errors introduced during the sampling process. Future research endeavors should consider incorporating metabolomics data in place of amplicon-based approaches. This shift would enable analyses to be conducted at the species level with enhanced accuracy, thereby improving the outcomes and enabling more precise conclusions”
